**Data Availability Statement:** All relevant data are within the manuscript and its Supporting Information files.

# Publication bias in trials registered in the Australian New Zealand Clinical Trials Registry: Is it a problem? A cross-sectional study

Marian Showell[1]*, Sam Buckman[1], Slavica Berber[2], Nada Ata Allah[3], Ben Patterson[4,5], Samantha Cole[1], Cynthia Farquhar[1], Vanessa Jordan[1]

1 Department of Obstetrics and Gynecology, University of Auckland, Auckland, New Zealand, 2 Evidence Integration, NHMRC Clinical Trials Centre, The University of Sydney, Camperdown, Australia, 3 Faculty of Medicine, Department of Obstetrics & Gynecology, Aleppo University Hospital, University of Aleppo, Aleppo, Syria, 4 Department of Medical Microbiology, St George's Hospital, London, United Kingdom, 5 Amsterdam Institute of Global Health and Development, University of Amsterdam, Amsterdam, The Netherlands

* m.showell@auckland.ac.nz

## Abstract

### Background

Timely publication of clinical trials is critical to ensure the dissemination and implementation of high-quality healthcare evidence. This study investigates the publication rate and time to publication of randomized controlled trials (RCTs) registered in the Australian New Zealand Clinical Trials Registry (ANZCTR).

### Materials and methods

We conducted a cross-sectional study of RCTs registered with the ANZCTR in 2007, 2009, and 2011. Multiple bibliographic databases were searched until October 2021 to identify trial publications. We then calculated publication rates, proportions, and the time to publish calculated from the date of first participation enrolment to publication date.

### Results

Of 1,970 trial registrations, 541 (27%) remained unpublished 10 to 14 years later, and the proportion of trials published decreased by 7% from 2007 to 2011. The average time to publish was 4.63 years. The prospective trial registration rate for 2007, 2009 and 2011 was 48% (952 trials) and over this time there was an increase of 19% (280 prospective trials). Trials funded by non-Industry organizations were more likely to be published (74%, 1204/1625 trials) than the industry-funded trials (61%, 224/345 trials). Larger trials with at least 1000 participants were published at a rate of 88% (85/97 trials) and on average took 5.4 years to be published. Smaller trials with less than 100 participants were published at a lower rate with 67% (687/1024 trials) published and these trials took 4.31 years on average to publish.

**Funding:** The author(s) received no specific funding for this work.

**Competing interests:** The authors have declared that no competing interests exist

## Conclusions

Just over a quarter of all trials on the ANZCTR for 2007, 2009, and 2011 remain unpublished over a decade later. The average time to publication of nearly five years may reflect the larger trials which will have taken longer to recruit participants. Over half of study sample trials were retrospectively registered, but prospective registration improved over time, highlighting the role of mandating trial registration.

## Introduction

We know that prompt reporting of clinical trials allows health information to be disseminated and implemented into practice and policy [1], however in reality for many trials this is not the case. Numerous studies have shown that the unpublished research rate remains high, approximately half of all existing clinical trials are not currently published [2–8]. The non-publication and delayed publication of research not only affects health care planning and delivery, but it is also a failure in our obligation to trial participants who have volunteered to participate in these trials with the expectation that their participation will help others [9–11]. In addition, there are safety concerns that an intervention's unknown, unpublished adverse effects may increase risks for future trial volunteers [12].

Publication bias occurs with the lack of accessibility of research results and is introduced when negative or null trial results are not published or their publication is delayed [6]. Studies are more likely to remain unpublished if they report negative findings, are funded by industry, or if the trials have small sample sizes [13–15]. We know that the quicker trial results are disseminated, the quicker new treatments are available to patients in need [16,17].

In an attempt to address the problem of lost or delayed health information, regulatory bodies such as the United States Food and Drug Administration (FDA), The Council of the European Union with the European Parliament, and the World Health Organisation all introduced legal means to censure trialists with fines of more than $10,000 a day for non-disclosure of results within one year of trial completion [12,18–21]. However, delayed publication and dissemination of results appear to remain a problem, with many studies publishing long after one year [14,22–26].

Alongside these regulatory requirements, there has been growing recognition of the need for trial registration that occurs before the first participant is enrolled, this is known as prospective trial registration, and any registration after this date is considered retrospective registration. Prospective trial registration provides publicly available evidence of a trial protocol, allowing for transparency from the date of first participant enrolment to the reported results [27]. As early as 1986, Simes [28] identified poor reporting of trials and called for establishing an international trial registry. In 2004 the International Committee of Medical Journal Editors (ICMJE) recommended that all clinical trials from 2005 onwards be registered before publication. They stated, 'All ICJME member journals will require registration in a public trials registry as a condition of consideration for publication' [29], and as of 2007, the requirement for trial registration had become standard practice for member journals [30]. Ethics committees and funding bodies are also trying to encourage registration and trial publication [31,32].

AllTrials, a global initiative launched in 2013, strongly advocated for the results of all clinical trials, past and present, to be registered and for these results to be publicly available within one year of trial completion [20]. The accessibility of RCTs on a trial registry allows researchers to report the extent of non-publication and publication bias [22]. However, prospective trial

registration rates continue to be concerning, with many multidisciplinary studies reporting low prospective trial registration rates and high retrospective registration rates [25,33–36]. Journals, funders, and ethics committees are asked to play a more significant role in increasing prospective trial registration [34].

National, regional, and international trial registries have emerged in response to the demand for public availability of registries. The Australian New Zealand Clinical Trials Registry (ANZCTR) is an online public register of clinical trials established in 2005 and, as of June 2022, contained 22,734 trials [37]. The ANZCTR aims to ensure trial information is "comprehensive, complete, current and compatible" [38]. It is one of 17 primary registries that feed into the World Health Organization International Clinical Trials Registry Platform (ICTRP), a global trials portal [37].

This study aims to assess research waste by determining the publication status and time to the publication of randomized controlled trials (RCTs) registered in the ANZCTR.

## Materials and methods

This study did not require ethics approval as the data was publicly available.

### Data source and study sample

This cross-sectional study assessed publication status and time to the publication of RCT's registered in the ANZCTR. We obtained a data set, from the ANZCTR of all publicly available data for each trial registered in 2007, 2009, and 2011. These years were selected to allow us to assess publication and the associated characteristics of trials over time. The year 2007 was an appropriate starting point for the study, as it was the year that the ANZCTR was first recognised as a Primary Registry by the World Health Organization International Clinical Trials Registry Platform (WHO ICTRP) [39] and it was also the year that prospective trial registration was mandated by the ICMJE member journals [30]. The final year of data analysis was 2011, this date allowed for an assessment of change over a five-year period, while allowing at least ten years for publication to occur. The midpoint year of 2009 was selected for practical reasons as it allowed us to monitor trends over time while keeping the number of trials assessed to a manageable level.

### Outcomes

We were interested in the following primary outcomes:

- The number and proportion of trial registrations that resulted in a publication;

- The time taken to publish these trials.

We also investigated whether the publication status and time to publication were associated with:

- A prospective or retrospective trial registration status;

- The country of origin for the primary author;

- Funding by industry or non-industry;

- The health condition studied;

- The target sample size of the trial;

- The study design (parallel, crossover, or factorial).

## Data collection

We counted a trial as published if we identified a published journal article, conference presentation with results, or a thesis. The three publication types were sub-grouped in the analyses.

We calculated the time to publication in years and months, from the date of anticipated first enrolment, found in the data of the ANZCTR download, to the publication date found in the paper. We used the earliest date provided for trials that published more than one paper or had different electronic or paper publication dates. We then calculated the mean time to publication.

The date of trial registration and the date of first participant enrolment was collected. A trial with a registration date after the first enrolment was considered retrospectively registered, while a registration before this was prospectively registered [27].

The country of the principal investigator determined the country of origin. For analysis, countries with ten or more trials were analyzed according to their country, while countries with less than ten associated trials were treated as one group and labeled 'Other.'

We checked the funding field (those designated to provide the funding or resources for the study) on the ANZCTR Website, and any funding gained from universities, hospitals, government, and charities was categorized as non-industry, while those trials with funding from the commercial sector were categorized as industry-funded. We then checked each non-industry-funded trial against the sponsorship field (those designated to take full legal responsibility for the study), and if a commercial sponsor was listed, these registrations were considered industry-funded.

Trials with less than 1000 participants were categorized into groups of 100, while trials with 1000 or more participants were considered in a single group.

Any trials that did not describe their RCT design type (parallel, cross over, or factorial) and trials coded as other were considered as one group, 'Unclear.'

## Search strategy to identify the publication of trials

Four investigators (MS, VJ, SB, and SC) independently determined the trial publication status by following a defined strategy. The last search was conducted on October 31, 2021, allowing a minimum time lag of ten years from registration to publication. We checked the ANZCTR registration number against Google Scholar. If the associated publication was not identified, we used other information described in the ANZCTR record to create a series of searches to match the trial. These included: public and scientific titles, inclusion/exclusion criteria, health condition, interventions and controls, gender/age of participants, the date of registration, and details from the principal investigator, including name, country, and institution.

Using this data, we began by searching MEDLINE and then progressed to Embase and The Cochrane Central Register of Controlled Trials and then Google (S1 Appendix).

When a potential trial was identified in the search, the full-text paper was checked for the ANZCTR registration number, and if the number was not reported in the paper, we confirmed the match through the other characteristics provided in the data set. Once a publication was confirmed, we collected the day, month, and year of publication. We used the first day of the month if no day was provided. A final step in the search process was to search the results field of the ANZCTR trial registration to find whether any of the unpublished trials had uploaded a summary of their results.

If a publication could not be identified after performing these searches, the trial registration was categorized as unpublished and, therefore, non-disseminated.

## Statistical analysis

The ANZCTR provided the trial registrations in an excel spreadsheet, and all data analyses were performed in Excel [40]. Dichotomous data was expressed in numbers and percentages. Continuous data was expressed as mean values and standard deviations (SD); if the data was not distributed normally, we used median values and interquartile ranges (IQRs). We tested for the significance of the primary outcomes; we used the Pearson's Chi-square test for dichotomous data, and for the continuous data of three or more sets, we used a single factor ANOVA test; we considered a $P$ value of $<0.05$ to be significant.

## Results

Of the 2,124 trials provided in the download from the ANZCTR, 1,970 were RCTs included in this study. We excluded 154 trials for being categorized as 'Withdrawn,' found to be a duplicate trial, or a non-randomized study.

## Publication

We searched the electronic bibliographic databases for publications matching the 1,970 ANZCTR registrations and found 1,429 (73%) trial registrations were published, of which 1,355 (95%) were journal publications, 56 (4%) were conference abstracts, and 18 (1%) were theses (Fig 1).

Only two registrations deposited results on the ANZCTR, having no other related publications. These were only brief texts with partial results and, for purposes of this research not considered adequate dissemination and were counted as unpublished.

Fig 2 reports the published and unpublished ANZCTR trial registrations for 2007 (467), 2009 (716), and 2011 (787).

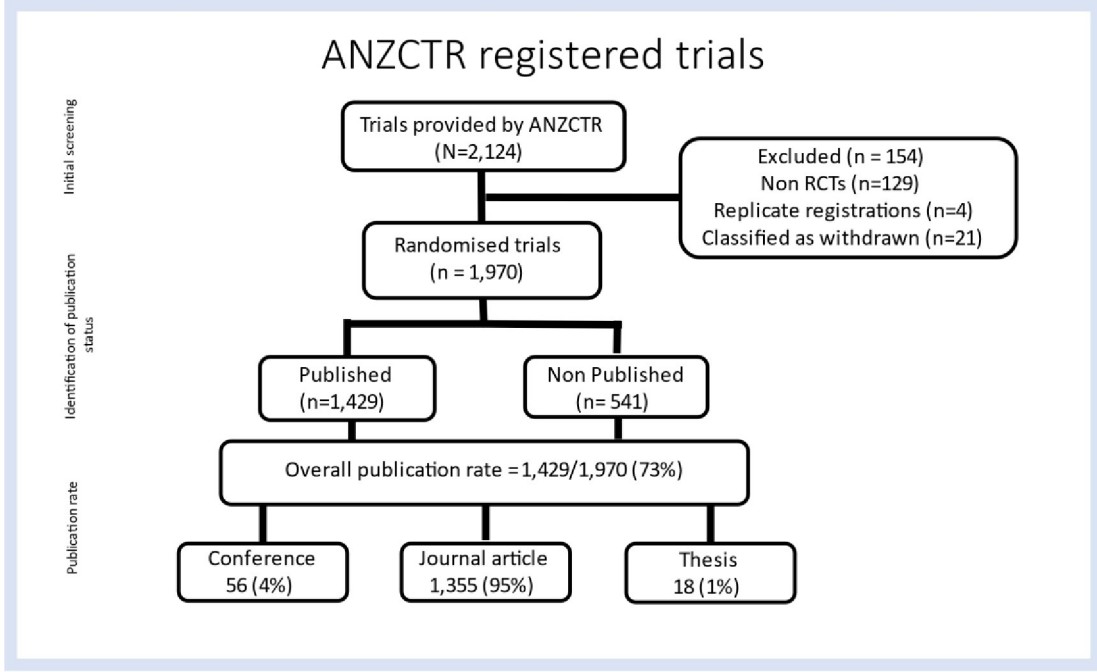

**Fig 1. Flow diagram from screening ANZCTR registrations to verification of publication status.**

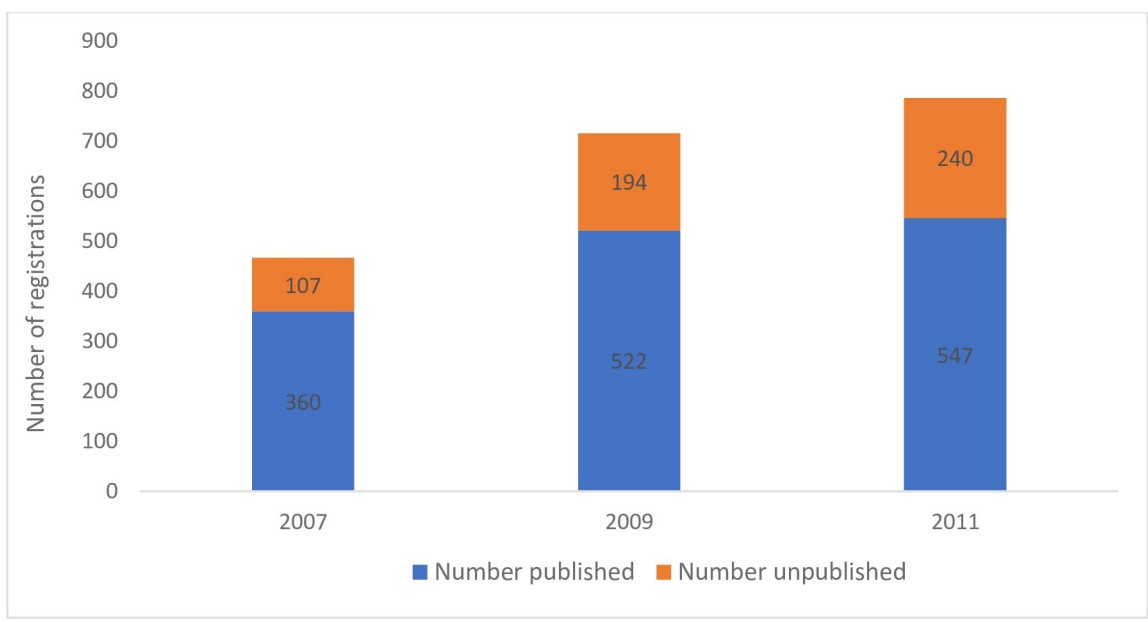

**Fig 2. Publication of ANTCTR registrations for the years 2007, 2009, and 2011 (n = 1,970).**

The proportions of published trials to registrations decreased over time with 360 (77%) in 2007, 522 (73%) 2009 and 547 (70%) in 2011, $X^2$ (2, $N = 1,970$) = 8.5, $p = 0.014$.

### Time to publication

The overall mean time to publication, from the date of first participant enrolment, was 4.63 years (SD 2.34). The conference abstracts, as expected, were quickest to publish with a mean time of 3.68 years (SD 2.66), then the theses with 3.95 years (SD 2.23). The longest time to publish was the full-text publications, with a mean of 4.68 years (SD 2.32). When comparing the different year groups of 2007 (m 5.16, SD 2.64), 2009 (m 4.67, SD 2.30) and 2011 (m 4.24, SD 2.10), the mean time to publication declined over these years (F (2,1425) = 17.18911, $p < .001$).

The majority of trials were published within five years from the date of first participant enrolment (Fig 3).

### Prospective or retrospective registration

Of the 1,970 trial registrations, 48% (952/1970) trials were registered prospectively and 52% (1,018/1970) were retrospectively registered. Of the 1,429 published trials, 46% (654/1429) were prospectively registered, and 54% (775/1429) were retrospective. Of the 541 unpublished trials, 55% (298/541) were prospectively registered, and 45% (243/541) were retrospectively registered. Prospective and retrospective trial registration in 2007, 2009, and 2011 are reported in Fig 4.

The proportion of publications decreased from 77% (2007) to 70% (2011), while prospective registration of published trials increased from 32% to 38%, and retrospective registration decreased from 45% in 2007 to 31% in 2011. Prospectively registered trials were published with a mean time of 4.68 years (SD 2.37), and retrospectively registered trials were published with an average of 4.59 years (SD 2.32).

### Primary country of author

Australia (71%) and New Zealand (11%) had the highest number of trials, with 71% and 79% publication rates, respectively. For countries with registered ten or more trials, Turkey

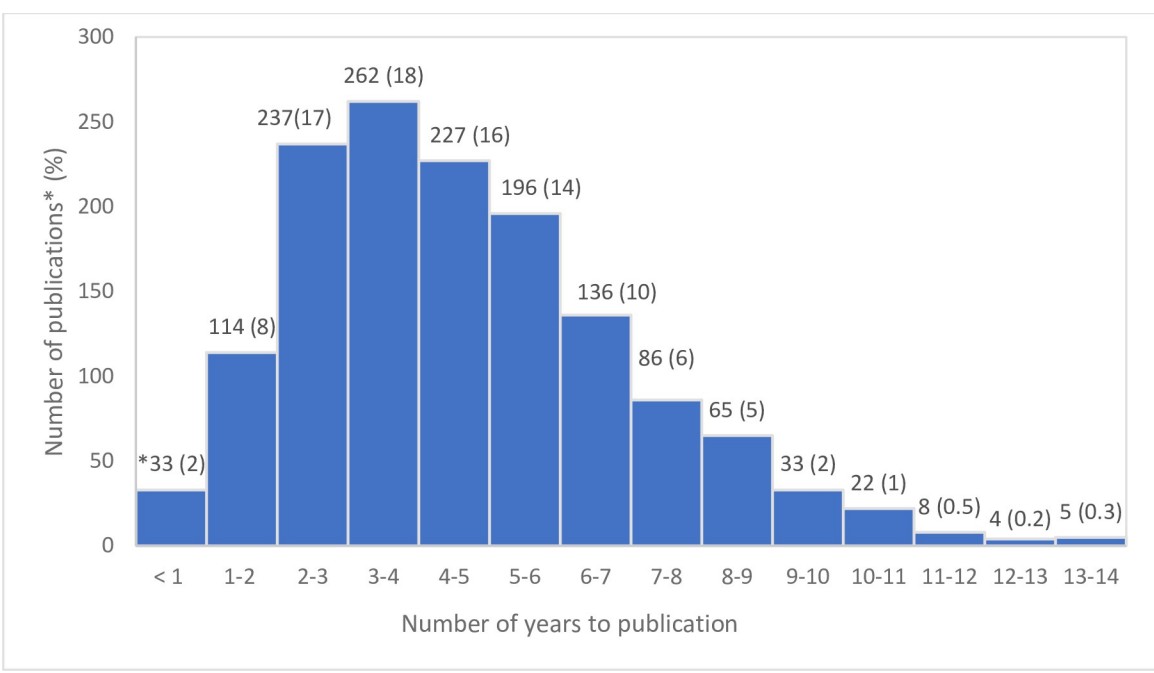

**Fig 3. Number of years from the date of first participant enrolment to publication, including prospective and retrospective registrations (n = 1,428).** * one conference abstract was not included in the analysis due to publication before the anticipated start date.

published the highest proportion with 91%, and Spain had the lowest publication rate with only four of the 12 trials (33%) published. All trial registrations from the countries with less than ten trials (Costa Rica, France, Papua New Guinea, each with one registered trial, and Austria, with seven trials) were unpublished.

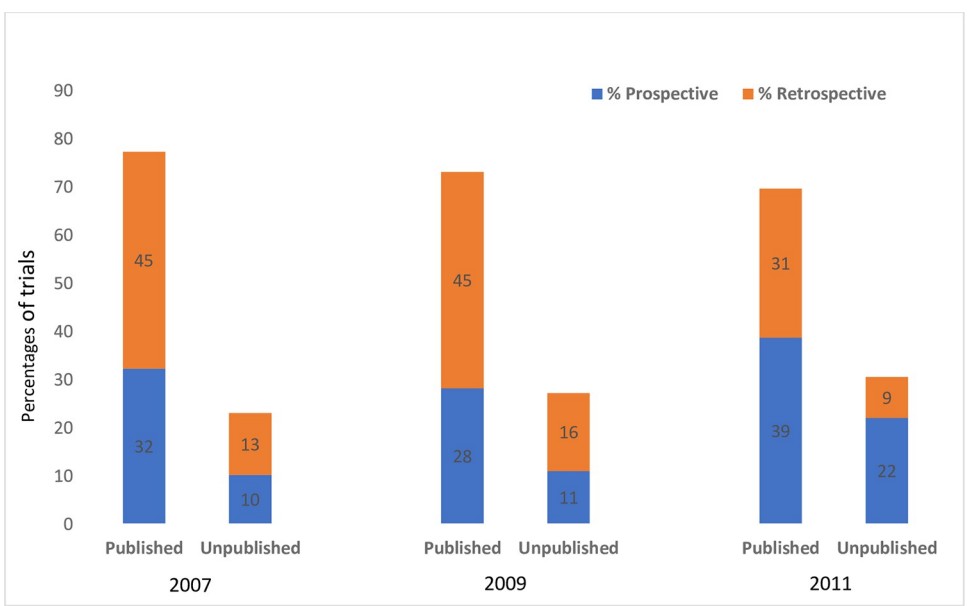

**Fig 4. Percentage of published and unpublished trials prospectively and retrospectively registered over time (n = 1,970).**

Of the countries with ten or more trials, Canada and The Republic of Korea shared the shortest time to publication, with a mean of 3.2 years (SD 2.25 and SD 1.83, respectively). The USA had the longest time to publication with a mean of 4.82 years (SD 1.87).

## Funding

Non-industry organizations funded 82% (1625/1970) of trials, and 74% were published. Of the 345 industry-funded trials, 65% were published.

The time to publication for industry or non-industry funded trials was a mean time of 4.72 (SD 2.27) and 4.61 (SD2.36) to publish, respectively.

Of the 345 industry-funded trials, 177 (51%) were prospectively registered, and of the 1,625 non- industry-funded trials, 776 (48%) were prospectively registered.

## Health condition

Trials in this dataset covered 26 different health conditions. The health condition with the highest number of registered trials was mental health, with 75% of trials under this health condition published. Ear conditions had the lowest proportion published (50%). However, there were only six trials in this cohort. Public Health was the category with the highest proportion of published trials (85%) (Fig 5).

When comparing time to publication for trials with different health conditions, we found that those looking at conditions of the ear had the shortest time to publication (m 3.03 years, SD 2.60), with only three trials in this group, and the longest were in human genetics and inherited disorders with an average time to publish of 5.91 years (SD 2.23).

## Sample size target

Results for the target sample size and subsequent publications are reported in Table 1.

The median target sample size of all trials was 90 (IQR 158). The median sample size for published trials was 100 (IQR 164) and 66 (IQR 92) for the unpublished trials.

Overall, the mean time to publication for all sample sizes ranged from 4.31 (SD 2.22) in the 0–99 sample size group to 5.4 (SD 2.72) in the trials with above 1000 participants. The trials with the larger participant numbers took longer to publish.

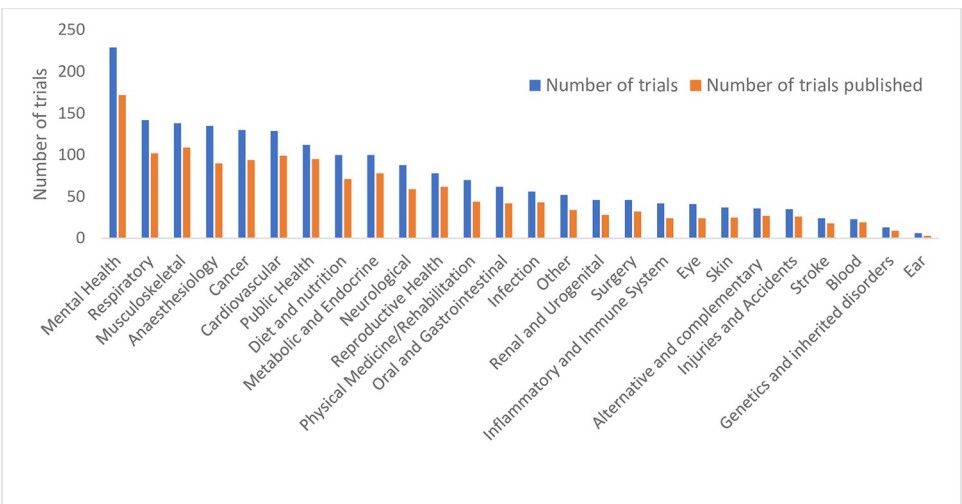

**Fig 5. Health conditions associated with ANZCTR trial registrations, number, and publication rate (n = 1,970).**

**Table 1. Target sample size and study design of ANZCTR registrations 2007, 2009, and 2011 (n = 1,970).**

| Target sample size | Number of Trials | Number of trials published (%) |
|---|---|---|
| 1–99 | 1024 | 687 (67) |
| 100–199 | 453 | 345 (76) |
| 200–299 | 158 | 119 (75) |
| 300–399 | 97 | 74 (76) |
| 400–499 | 42 | 37 (88) |
| 500–599 | 25 | 20 (80) |
| 600–699 | 38 | 30 (79) |
| 700–799 | 15 | 13 (87) |
| 800–899 | 15 | 13 (87) |
| 900–999 | 6 | 6 (100) |
| >/ = 1000 | 97 | 85 (87) |
| **Total** | **1970** | 1429 |
| **Study design** | | |
| Parallel | 1518 | 1112(73) |
| Crossover | 242 | 172 (71) |
| Factorial | 26 | 21 (81) |
| Unclear | 184 | 124 (67) |
| **Total** | **1970** | 1429 |

Smaller trials were more likely to be retrospectively registered in the 0–99 sample size group (54%), while trials with over 1000 participants were less likely to be retrospectively registered (42%).

## Study design

The majority of RCTs used parallel design 1,518 (77%), followed by crossover design, but the factorial design trials had the highest publication rate with 21 publications (81%) (Table 1).

Crossover design trials were published in a mean of 4.11 years (SD 2.28), factorial design trials in 4.29 years (SD 2.39), and parallel design trials took the longest to publish with a mean of 4.72 years (SD 2.37).

## Discussion

### Main findings

Twenty-seven percent of ANZCTR trials remained unpublished 10 to 14 years after trial registration, representing the loss of results from 541 trials and the data from 144,435 trial participants. The average time to publish decreased from just over five years (2007) to over four years (2009, 2011). In the 2007 cohort, nearly 60% of trials were registered retrospectively; this decreased to 40% by 2011, increasing prospective registration.

Our study has shown that non-industry funded trials were more likely to be published than industry-funded and that there was no real difference in time taken to publish between the industry and non-industry trials. The health condition with the highest publication rate was public health. Larger trials had higher publication and prospective registration rates, with a longer time to publish than smaller trials.

### Strengths and limitations

As far as we know, this is the first study looking at publication status and time to the publication of ANZCTR trials. We aimed to assess how the regulatory trial requirements introduced in 2005 and mandated in 2007 have been met in the Australasian region.

A major strength of our study was the systematic and rigorous approach to identifying publications over multiple bibliographic databases. Other strengths were the inclusion of alternative methods of publication with the addition of conference abstracts and theses, allowing for an exploration of different avenues of disseminating results [13,21]. We also searched for the publications ten years after the 2011 cohort date of first participant enrolment, allowing more than sufficient time for trial publication [13].

The major limitation of the study was that we did not identify the date of the last patient enrolment and report this time to publication, we instead reported on the anticipated date of first patient enrolment and publication, and in doing so, we may have missed helpful information on reasons for delayed publication. The main reason for this is that the majority of studies did not report the date of last participant enrolment.

Although our search for publications was detailed and extensive and widened by including other types of reporting, i.e., conference abstracts and theses, we accept that we may not have found some publications and that personal contact with trial investigators may have mitigated this. However, we believe the number of potentially missed trials would be small, and if those obscure and difficult-to-find publications exist, we think that, although they would legally fulfill publication requirements, they would not adequately meet the ethical requirement of easy-to-find trial results.

## Interpretation

Publication bias remains a problem for ANZCTR registered trials in 2007, 2009, and 2011. While regulatory bodies mandated disclosure of results within one year of trial completion, our study found that the number of publications increased. However, the proportion of publications to registrations had decreased, and just over a quarter of all trials remain unpublished. This result is similar to the publication results in many other studies. Ross [14] found that one-third of the US National Institutes of Health funded trials registered from 2005 to 2010 on clinicaltrials.gov remained unpublished after four years. A study [26] of Australian-funded trials between 2008 and 2010 found that nearly half were not published. While other studies looking at trial registries also found similar results, an analysis of the Trial Register of The Netherlands database [8] found that one out of four trials remained unpublished after five years, and 40% of trial registrations were retrospective. Our study included more recent trial registrations than these earlier studies (including trial registrations from 2011); however, our results showed similar proportions of reported results.

We accept that there are many barriers to publication [41], but very few of the unpublished trials took the opportunity to upload summary results onto the ANZCTR portal or to release conference abstracts or theses. However, we found an improvement in the average time taken to publish, with a decrease from five years (2007) to four years (2011) from the first participant enrolment.

The proportion of prospective registration of published and unpublished studies also improved but remained low, with only 48% of the total number of trials prospectively registered. The unpublished trials had a higher number of prospectively registered trials than the published. Over time we have seen prospective registration rise as trial registration becomes more prevalent.

Prospectively registered trials were published with a mean time of 4.68 years (SD 2.37), and retrospectively registered trials were published with an average of 4.59 years (SD 2.32). We had expected a greater difference here, as we had assumed that prospectively registered trials would take much longer to publish than retrospective trials due to the assumption that many are registered just before publication to meet the journal's mandatory requirement for registration prepublication. Perhaps, this similarity in time to publish can be explained in part by

retrospective registration that occurs just after the cut-off date of first-person enrolment. So, the retrospective cohort is not only made up of trials registered at the point of publication but also includes those registered earlier in the trial process. A study by Farquhar [34] shows this; they studied three different registration times; before the first enrolment, one to six months after enrolment, and anything after the six months of the first enrolment, and found that 16% of these trials were registered in the 6-month time-period prior to data collection. The methods in these trials could not be manipulated as a result of their findings, so perhaps when performing studies like ours, there is a need to treat this group differently from those trials registered later in the trial process.

It remains a problem that 27% of publicly funded trials remain unpublished, although these trials were still more likely to be published than those funded by industry. Yilmaz confirmed this [42] by looking at trials enrolling participants with cognitive impairment and found that the odds of non-publication in industry-sponsored trials were 75% higher than in studies funded by academia. In contrast, a study by Ross [43] and a recently published systematic review found no difference in publication rate between the two funding types [44].

The ANZCTR reported [45] that the prospective registration rate of ANZCTR registered trials in 2012 was 67% and "has since plateaued at around 65–70 percent". Our study showed a prospective registration rate of 48% for the years 2007 to 2011; this is an 18% increase, while in the same time period, the proportion of published trials to registrations decreased by 7%. This change may reflect the regulatory requirement to register the trial before enrolling the first participant [27], which only came into effect after 2007 [29,46]. The increased proportion of prospectively registered trials may, in turn, create a longer time interval between the first participant enrolment date and potential publication date. A prospectively registered trial would have more hurdles to overcome, and many trials may never start or may be terminated early [16,17]. If these problems cannot be solved, the trial will be identifiable in the register but not be published, resulting in a lower publication to registration rate. Whereas a retrospective registration closer to the time of publication may have already overcome these issues and will go ahead and be published, retrospective trials that do not overcome these barriers will disappear with no public knowledge of them ever existing. This action ignores our obligations to trial participants [9,12].

New international and national initiatives are underway that aim to decrease publication bias and improve trial quality. Internationally, The SPIRIT guidelines [47] and the SPIRIT Electronic Protocol & Resource (SEPTRE) [48], a web-based tool introduced in 2020, are helping trialists improve the content and quality of protocols while assisting with the management and prospective registration of trial protocols. At a national level, The Health Research Authority (HRA) in the United Kingdom has announced that from 2022, trials will be automatically registered at the point of ethics approval [49], thus ensuring research transparency by taking the responsibilities of trial registration from the individual trialists. While in Australia, The Australian Commission on Safety and Quality in Health Care is seeking consultation with health stakeholders to develop 'The National One Stop Shop' [50]. As the name suggests, this will be a central point of access for all aspects of the life of a trial; these include trial and data management, ethics approval, trial registration, monitoring and reporting, and funding. This platform aims to support trialists, improve trial quality, and report results by centralizing and coordinating trial activity. Other countries could do well to follow these initiatives.

## Conclusions

Despite an improvement in time to publication for trials registered in the ANZCTR in the years 2007, 2009, and 2011, we found that the proportion of trial publications to registrations

decreased, and 27% of registrations remain unpublished up to 14 years after the first participant enrolment. All trial stakeholders now need to act collaboratively to ensure the timely dissemination of trial results into research evidence and clinical practice.

## Supporting information

**S1 Appendix. Search strategy algorithm.**
(TIF)

**S1 Data.**
(XLSX)

## Acknowledgments

We would like to acknowledge the ANZCTR for providing us with this data, with a special mention to Melina Willson who not only assisted with the transfer of this data but also provided in depth comment on the final draft of the paper.

## Author Contributions

**Conceptualization:** Marian Showell, Sam Buckman, Cynthia Farquhar, Vanessa Jordan.

**Data curation:** Marian Showell, Sam Buckman, Nada Ata Allah, Ben Patterson, Samantha Cole, Vanessa Jordan.

**Formal analysis:** Marian Showell.

**Investigation:** Marian Showell, Sam Buckman, Vanessa Jordan.

**Methodology:** Marian Showell, Sam Buckman, Vanessa Jordan.

**Project administration:** Marian Showell, Vanessa Jordan.

**Resources:** Marian Showell, Slavica Berber.

**Software:** Marian Showell.

**Supervision:** Marian Showell, Cynthia Farquhar, Vanessa Jordan.

**Validation:** Marian Showell.

**Visualization:** Marian Showell, Vanessa Jordan.

**Writing – original draft:** Marian Showell.

**Writing – review & editing:** Marian Showell, Sam Buckman, Slavica Berber, Nada Ata Allah, Ben Patterson, Samantha Cole, Cynthia Farquhar, Vanessa Jordan.

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
