## [Decision Letter · Decision Letter 0]

11 Nov 2022

PONE-D-22-24255Publication bias in trials registered in the Australian New Zealand Clinical Trials Registry: Is it a problem? A cross-sectional studyPLOS ONE

Dear Dr. Showell,

Thank you for submitting your manuscript to PLOS ONE. After careful consideration, we feel that it has merit but does not fully meet PLOS ONE’s publication criteria as it currently stands. Therefore, we invite you to submit a revised version of the manuscript that addresses the points raised during the review process.

We look forward to receiving your revised manuscript.

Kind regards,

Nabeel Al-Yateem, PhD

Academic Editor

PLOS ONE

Journal Requirements:

Reviewers' comments:

Reviewer's Responses to Questions

**Comments to the Author**

1. Is the manuscript technically sound, and do the data support the conclusions?

Reviewer #1: Yes

2. Has the statistical analysis been performed appropriately and rigorously? 

Reviewer #1: Yes

3. Have the authors made all data underlying the findings in their manuscript fully available?

Reviewer #1: Yes

4. Is the manuscript presented in an intelligible fashion and written in standard English?

Reviewer #1: Yes

5. Review Comments to the Author

Reviewer #1: An important piece of work on the dissemination status of registered trials at the ANZCTR. The methodology is sound and the report clear and easy to follow. I have only the following minor suggestions:

1. Introduction: lines 69-70: " We know that the quicker trial results are disseminated, the more clinically relevant these results become (16, 17)" - the authors may wish to clarify what is meant by the term "clinically relevant" when used in this context - are the results more likely to be incorporated in clinical decision making or practice guideline, lead to practice change, or the authors meant something else?

2. Introduction, lines 102-103: the authors may wish to explain here specifically why these three particular years (2007, 2009 and 2011) were selected for this study.

3. Data collection, lines 124-125: "We calculated the time to publication in years and months, from the date of anticipated first enrolment, found in the data of the ANZCTR download, to the publication date found in the paper." - what about retrospectively registered trial? Which date was taken as the first point of measurement?

4. Data collection, lines 142-143: "Any trials that did not describe their RCT design type (parallel, cross over, or factorial) and trials coded as other were considered as one group, 'Other.'" - should they instead be labelled as "unclear" as there was a lack of required information?

5. Search Strategy to identify the publication of trials - it will be very helpful for future researchers who wish to do similar work if a generic methodology or algorithm in tracing publication used in this study, if available, be made available either in the main text or in the appendix. This will set a standard for others to follow and possibly update as required.

The rest of the manuscript is well-written and comprehensively reported.

6. PLOS authors have the option to publish the peer review history of their article (what does this mean?). If published, this will include your full peer review and any attached files.

Reviewer #1: **Yes: **Nai Ming Lai

---

## [Author Response · Author response to Decision Letter 0]

16 Nov 2022

Reply to the comments from the Journal Editor

Comment 1: “Please ensure that your manuscript meets PLOS ONE's style requirements, including those for file naming 

Response: I have followed the guidance provided regarding PLOS ONE’s style requirements and made changes as necessary.

Comment 2: “In your Data Availability statement, you have not specified where the minimal data set underlying the results described in your manuscript can be found. PLOS defines a study's minimal data set as the underlying data used to reach the conclusions drawn in the manuscript and any additional data required to replicate the reported study findings in their entirety. All PLOS journals require that the minimal data set be made fully available”.

Response: I have uploaded our underlying data det as a Supporting Information file S2 (excel spreadsheet) to PLOS ONE 

Comment 3: “We note that you have stated that you will provide repository information for your data at acceptance”

Response: I have now decided to upload the data to PLOS ONE so the repository information is no longer required. I have described these changes in the cover letter.

Comment 4: “Please review your reference list to ensure that it is complete and correct”

Response: I have reviewed the reference list and no changes were made.

Reply to the comments from the reviewers

Reviewer one

Introduction

Comment 1: “lines 69-70: " We know that the quicker trial results are disseminated, the more clinically relevant these results become (16, 17)" - the authors may wish to clarify what is meant by the term "clinically relevant" when used in this context - are the results more likely to be incorporated in clinical decision making or practice guideline, lead to practice change, or the authors meant something else?”

Response: I have reworded this sentence to clearly explain what we meant by “clinically relevant”

Comment 2: “lines 102-103: the authors may wish to explain here specifically why these three particular years (2007, 2009 and 2011) were selected for this study.”

Response: I have added an explanation for the selected years.

Data collection

Comment 3: lines 124-125: "We calculated the time to publication in years and months, from the date of anticipated first enrolment, found in the data of the ANZCTR download, to the publication date found in the paper." - what about retrospectively registered trial? Which date was taken as the first point of measurement?

Response: Both prospectively and retrospectively registered trials provided a date of first participant enrolment which was taken as the first point of measurement.

Comment 4: “lines 142-143: "Any trials that did not describe their RCT design type (parallel, cross over, or factorial) and trials coded as other were considered as one group, 'Other.'" - should they instead be labelled as "unclear" as there was a lack of required information?”

Response: I have now changed this category from “Other” to “Unclear”

Comment 5: Search Strategy to identify the publication of trials - it will be very helpful for future researchers who wish to do similar work if a generic methodology or algorithm in tracing publication used in this study, if available, be made available either in the main text or in the appendix. This will set a standard for others to follow and possibly update as required.

Response: Thank-you for this suggestion. I have now created a pdf of the search algorithm and uploaded it to PLOS ONE as a supplementary file (S1_Appendix). 

Comment 6: “PLOS authors have the option to publish the peer review history of their article. If published, this will include your full peer review and any attached files.”

Response: Yes

---

## [Decision Letter · Decision Letter 1]

12 Dec 2022

PONE-D-22-24255R1Publication bias in trials registered in the Australian New Zealand Clinical Trials Registry: Is it a problem? A cross-sectional studyPLOS ONE

Dear Dr. Showell,

Thank you for submitting your manuscript to PLOS ONE. After careful consideration, we feel that it has merit but does not fully meet PLOS ONE’s publication criteria as it currently stands. Therefore, we invite you to submit a revised version of the manuscript that addresses the points raised during the review process.

We look forward to receiving your revised manuscript.

Kind regards,

Nabeel Al-Yateem, PhD

Academic Editor

PLOS ONE

Journal Requirements:

Reviewers' comments:

Reviewer's Responses to Questions

**Comments to the Author**

1. If the authors have adequately addressed your comments raised in a previous round of review and you feel that this manuscript is now acceptable for publication, you may indicate that here to bypass the “Comments to the Author” section, enter your conflict of interest statement in the “Confidential to Editor” section, and submit your "Accept" recommendation.

Reviewer #1: All comments have been addressed

2. Is the manuscript technically sound, and do the data support the conclusions?

Reviewer #1: Yes

3. Has the statistical analysis been performed appropriately and rigorously? 

Reviewer #1: Yes

4. Have the authors made all data underlying the findings in their manuscript fully available?

Reviewer #1: Yes

5. Is the manuscript presented in an intelligible fashion and written in standard English?

Reviewer #1: Yes

6. Review Comments to the Author

Reviewer #1: Thank you for attending to the comments. There remains one point that I would like clarification. In my comment no 2: “lines 102-103: the authors may wish to explain here specifically why these three particular years (2007, 2009 and 2011) were selected for this study.” As far as I can see, the author justified the selection of the years by stating that they wished to allow at least 10 years to assess publication status. That is fine, my point was why picked 2007, 2009 and 2011, i.e. and not including years like 2008 and 2010. I don't seem to have seen any explanation on this. Apologies if I have missed something obvious. All the best.

7. PLOS authors have the option to publish the peer review history of their article (what does this mean?). If published, this will include your full peer review and any attached files.

Reviewer #1: **Yes: **Nai Ming Lai

---

## [Author Response · Author response to Decision Letter 1]

15 Dec 2022

Thank you for clarifying your comment regarding the need for a clearer explanation of why we chose to study the years 2007, 2009 and 2011. We have added this explanation to the methods section. You will see that 2007 is chosen as the year when trial registration become mandatory and the ANZCTR became a recognised registry by the WHO, the year 2011 allowed for at least ten years to publication and we chose to use 2009 as the mid point as we were unable to cope in practical terms with five years of data from the years 2007, 2008, 2009, 2010 and 2011.

---

## [Editor Report · Decision Letter 2]

19 Dec 2022

Publication bias in trials registered in the Australian New Zealand Clinical Trials Registry: Is it a problem? A cross-sectional study

PONE-D-22-24255R2

Dear Dr. Showell,

We’re pleased to inform you that your manuscript has been judged scientifically suitable for publication and will be formally accepted for publication once it meets all outstanding technical requirements.

Kind regards,

Nabeel Al-Yateem, PhD

Academic Editor

PLOS ONE
---

## [Editor Report · Acceptance letter]

22 Dec 2022

PONE-D-22-24255R2 

Publication bias in trials registered in the Australian New Zealand Clinical Trials Registry: Is it a problem? A cross-sectional study 

Dear Dr. Showell:

I'm pleased to inform you that your manuscript has been deemed suitable for publication in PLOS ONE. Congratulations! Your manuscript is now with our production department. 

Kind regards, 

on behalf of

Dr. Nabeel Al-Yateem 

Academic Editor

PLOS ONE